# Major Contribution of c.[1622T>C;3113C>T] Complex Allele and c.5882G>A Variant in *ABCA4*-Related Retinal Dystrophy in an Eastern European Population

**DOI:** 10.3390/ijms242216231

**Published:** 2023-11-12

**Authors:** Vitaly V. Kadyshev, Ekaterina A. Alekseeva, Vladimir V. Strelnikov, Anna A. Stepanova, Alexander V. Polyakov, Andrey V. Marakhonov, Sergey I. Kutsev, Rena A. Zinchenko

**Affiliations:** Research Centre for Medical Genetics, 115522 Moscow, Russia; ekater.alekseeva@gmail.com (E.A.A.); vstrel@list.ru (V.V.S.); anna_stepanova@med-gen.ru (A.A.S.); polyakov@med-gen.ru (A.V.P.); marakhonov@generesearch.ru (A.V.M.); kutsev@mail.ru (S.I.K.); renazinchenko@mail.ru (R.A.Z.)

**Keywords:** *ABCA4*, Stargardt disease, cone-rode dystrophy, age-related macular dystrophy, retinitis pigmentosa, complex allele, genotype–phenotype correlation

## Abstract

Inherited retinal diseases (IRDs) constitute a prevalent group of inherited ocular disorders characterized by marked genetic diversity alongside moderate clinical variability. Among these, *ABCA4*-related eye pathology stands as a prominent form affecting the retina. In this study, we conducted an in-depth analysis of 96 patients harboring *ABCA4* variants in the European part of Russia. Notably, the complex allele c.[1622T>C;3113C>T] (p.Leu541Pro;Ala1038Val, or L541P;A1038V) and the variant c.5882G>A (p.Gly1961Glu or G1961E) emerged as primary contributors to this ocular pathology within this population. Additionally, we elucidated distinct disease progression characteristics associated with the G1961E variant. Furthermore, our investigation revealed that patients with loss-of-function variants in *ABCA4* were more inclined to develop phenotypes distinct from Stargardt disease. These findings provide crucial insights into the genetic and clinical landscape of *ABCA4*-related retinal dystrophies in this specific population.

## 1. Introduction

Inherited retinal diseases (IRDs) represent a diverse group of genetic eye disorders that exhibit substantial genetic heterogeneity alongside moderate clinical variability [1]. Autosomal recessive inheritance accounts for 50–60% of IRD cases, autosomal dominant inheritance for 30–40%, and X-linked inheritance for 5–15% [2]. Currently, over 280 genes have been linked to IRD [3,4]. The classification of IRD is primarily based on retinal lesion topography (central, peripheral, mixed forms) and the site of pathological involvement (retinal pigment epithelium (RPE) lesions, “pigment epithelium—photoreceptors” complex, photoreceptors, inner retinal layer, Bruch’s membrane, “retina—vitreous body” complex, choroidal vessels, and atypical fundus presentations) [5]. Nonetheless, these classifications only partially capture etiopathogenetic mechanisms and molecular underpinnings of the disorders.

This publication is dedicated to exploring retinal pathology associated with the *ABCA4* gene. Genetic mutations within the *ABCA4* gene give rise to four distinct clinical and genetic forms of retinal disorders: Stargardt disease type 1 (STGD1, OMIM #248200), cone-rod dystrophy type 3 (CORD3, OMIM #604116), retinitis pigmentosa type 19 (RP19, OMIM #601718), and age-related macular dystrophy type 2 (ARMD2, OMIM #153800). The *ABCA4* gene, encompassing 50 exons, encodes a protein comprising 2773 amino-acid residues. This protein is primarily expressed in the outer segments of photoreceptor cells and plays a pivotal role in metabolizing visual cycle intermediates. Dysfunctional ABCA4 protein precipitates cytotoxicity targeting the retinal pigment epithelium (RPE), culminating in RPE and photoreceptor cell dysfunction and demise [6].

Presently, about 2000 genetic variants in this gene have been documented, exhibiting diversity across global populations according to HGMD v.2022.1. This diversity encompasses distinct complex alleles and phenotypic variability in different genotypes [7,8,9]. While some studies have explored genotype–phenotype correlations, statistically robust links between genotypes and phenotypes remain considerably constrained. This study aimed to explore the genetic diversity within *ABCA4* allelic variations among patients displaying different forms of retinal lesions. Additionally, the objective was to evaluate potential correlations between the clinical presentations and genotypes of the identified nosologic forms in the Russian Federation.

## 2. Results

### 2.1. Genetic Heterogeneity in the ABCA4 Gene

A total of 72 distinct genetic variants impacting the *ABCA4* gene have been discerned. Among these, there are 46 distinct missense substitutions and one known synonymous variant (c.4506C>T, p.(Cys1502=)). Additionally, there are variants classified as loss-of-function (LoF) alterations, encompassing one gross deletion, three frame-shifting variants, eight nonsense substitutions, and five variants influencing canonical splicing sites. Furthermore, there are eight additional variants situated outside the canonical splice site ± 2 bp region (see Figure 1, Appendix A). Among the patients, three individuals were found to possess only one identified variant in their genotype. Meanwhile, 48 patients (50%) exhibited more than two variants within their genotype, leading to the emergence of complex alleles. Among this group, the most frequently observed complex allele was c.[1622T>C;3113C>T], p.[(Leu541Pro;Ala1038Val)], which was detected in 37 patients. The most common third allele in trans with this complex allele was c.5882G>A, p.(Gly1961Glu), a variant found in 11 unrelated patients. In 10 patients, two variants, c.1622T>C and c.3113C>T, were located in trans. In a subset of 23 patients, loss-of-function (LoF) variants were detected in at least one allele (Appendix A).

### 2.2. Clinical Heterogeneity in the ABCA4 Gene

The study included a mandatory retrospective study of medical records and data of instrumental examination of the patient at an earlier (childhood) age; as a result, the registered data on the disease debut, rather than the age of the patient at the time of the present study, were taken into account when assessing the criterion of the age of manifestation. This was necessarily taken into account in the statistical analysis. Indeed, the older the patient becomes, the more severe the disease is; that is why the data of instrumental examination (optical coherence tomography, perimetry, visual acuity, electroretinography data) in dynamics were analyzed. This is an extremely important component of the study, as it is incorrect to establish the age of the pathology debut only on the basis of the data obtained at the time of this study. Taking into account the localization of *ABCA4* gene expression and clinical variability, the pathological process may debut from both pigment epithelium and photoreceptor discs, or it may debut simultaneously from both components, due to which a different clinical portrait develops and a different clinical diagnosis is formed. In view of these facts, different degrees of damage in certain retinal layers (cone and rod apparatus, pigment epithelium, outer boundary membrane) are recorded and, therefore, corresponding changes in functional vision parameters are registered, including color perception and maximum corrected visual acuity.

Stargardt’s disease type 1 (STGD1) constitutes a significant portion of the confirmed inherited retinal diseases (IRDs) spectrum, amounting to 88.5% (85 out of 96 cases). Within the STGD1 spectrum, there exists a pronounced genetic and clinical heterogeneity. The age at which symptoms initiate spans from 7 to 15 years, though rare patients have shown onset beyond the age of 50. The severity of the phenotype also exhibits substantial variation (see Appendix A).

The most frequent cause of STGD1 within our cohort is the complex allele c.[1622T>C;3113C>T], p.[(Leu541Pro;Ala1038Val)], identified in 34 cases, accounting for 40.0% of patients.

Typically, STGD1 manifests as a classical clinical profile. Ophthalmoscopic examination commonly unveils a distinctive macular region characterized by atrophic changes displaying a metallic sheen; notable is the absence of observed pigment deposits. Notably, the atrophic alterations are limited to the paramacular region, without extending further. These structural observations are in alignment with the findings from electroretinography (Figure 2).

Retinitis pigmentosa type 19 (RP19) was identified in five cases, accounting for 4.17% of the total (5 out of 96). In all cases of RP19, a distinct pattern emerged that was characterized by a comprehensive atrophic alteration of the retinal pigment epithelium (RPE), juxtaposed with a relatively conserved macular region structure (Figure 3a,b). Among patients aged above 20 years, involvement of the photoreceptor layer within the pathological process was evident; however, this layer could still be discerned. Across all individuals affected by this condition, central vision demonstrated a prolonged preservation phase, even in the presence of a pronounced constriction of peripheral visual fields. Moderate pigment deposition, specifically of the “bone bodies” type, was notable on fundus examination (Figure 3c,d). It’s important to highlight that the optic disc, within this clinical subtype, exhibited partial indications of an atrophic process.

Cone-rod dystrophy type 3 (CORD3) was identified in four patients exhibiting alterations in the *ABCA4* gene. This condition was marked by a profound suppression of dark adaptation along with compromised color perception. Notably, optical coherence tomography unveiled conspicuous reduction in photoreceptor height, while the retinal pigment epithelium layer exhibited sustained integrity over an extended duration. The onset of the pathological process in CORD3 was specifically localized to the photoreceptor layer (Figure 4). Importantly, there were no patients with recorded pigment deposition on the ocular fundus.

Age-related macular dystrophy type 2 (ARMD2) was detected in two patients. Onset of this condition was observed after 23–24 years of age, marked by a progressive decline in central vision and diminished color perception. It is worth highlighting that the pathological progression encompassed nearly all sectors of the outer retina. Notably, neovascularization was absent in all patients with this condition.

### 2.3. Genotype–Phenotype Correlations in ABCA4-Related IRD

The most prevalent group of patients in our cohort was patients with STGD1. Since the most common allele in this group was complex allele c.[1622T>C;3113C>T], which was found in 34 patients, we compared the clinical picture of this variant with that observed in a group of compound-heterozygous probands having a c.[1622T>C];[3113C>T] genotype (n = 10). This comparison did not reveal statistically significant associations (*p*-value > 0.05).

We next focused on the STGD1 patients with the c.[1622T>C;3113C>T];[5882G>A] (n = 11) genotype, since this genotype was the most prevalent in the group of patients with complex alleles. We compared the clinical picture of patients with this genotype with those who had the c.[1622T>C];[3113C>T] compound-heterozygous genotype (n = 10). Statistical analysis revealed that patients with the c.[1622T>C;3113C>T];[5882G>A] genotype demonstrated a significantly higher degree of disease manifestation which corresponds to later onset of the disease (*p*-value = 0.0176, Freeman–Halton extension of the Fisher exact probability test).

Since a substantial part of our cohort represents patients with so called loss-of-function (LoF) alleles, we next focused on the clinical diagnosis of patients with this type of variant in genotype. Other than STGD1, diagnosis was significantly more frequently associated with presence of LoF variant in genotype (Table 1; *p* = 0.0210, Fisher exact probability test). The odds ratio of development of STGD1 in a patient with the LoF variant in genotype was 0.21 (95% confidence interval 0.05–0.95).

## 3. Discussion

Inherited retinal diseases are the most widespread and numerous group of hereditary ophthalmic pathologies, characterized by a wide genetic heterogeneity (locus and allele) and a variety of clinical and genetic forms with similar phenotypic manifestations.

We presented the results of a multifaceted clinical and genetic analysis performed on a cohort of 96 patients with verified *ABCA4*-related hereditary eye pathology from the Russian population. A total of 72 distinct genetic variants in the *ABCA4* gene were identified. Among them, a separate group represents loss of function variants (defined according to molecular consequence).

A total of 37 patients (38.54%) possessed the pathogenic complex allele in *ABCA4*—c.[1622T>C;3113C>T], the allelic frequency of which appeared to be 19.2%. The presence of complex alleles has been found for many genes and can affect the clinical manifestations of diseases, both aggravating the clinical picture and leading to a milder course of the disease. No patients were revealed with this allele in a homozygous state. One of the most intriguing groups of patients was those with the compound heterozygous genotype c.[1622T>C];[3113C>T]. To the best of our knowledge, this is the first report of these two variants separated on different chromosomes, since previous data suggest that they are always located on one allele [10,11].

The c.[1622T>C;3113C>T] variant is the main complex allele occurring in the cis position in *ABCA4*-associated posterior eye pathology. In most cases, researchers provide clinical examples of one or more cases.

In a study by Fishman G. A. et al., it was shown that patients who were homozygous for the p.[L541P;A1038V] complex allele exhibited a variety of clinical symptoms, including retinitis pigmentosa, cone-rod dystrophy, age-related macular degeneration, and Stargardt’s disease. They presented a clinical case of a 35-year-old woman of Polish-German origin, homozygous for the complex allele c.[1622T>C;3113C>T], who was diagnosed with cone-rod dystrophy [7]. The patient’s visual acuity was significantly reduced (OD 10/350 and OS 5/400), and loss of the central and peripheral visual fields was observed. An ERG showed the same reduction in the number of cones and rods, and ocular fundus findings indicated diffuse pigmentary degenerative changes.

A clinical case of a family from Latvia with two patients with retinitis pigmentosa caused by a mutation in the *ABCA4* gene with the c.[1622T>C;3113C>T] complex allele in the homozygous state was described [3]. Homozygosity of the c.[1622T>C;3113C>T] complex allele has been shown to cause a severe phenotype characterized by early manifestation and retinal and macular lesions [12]. This complex is more frequent (12.7% of patients with Stargardt’s disease) in people of German origin [13], compared to 1.1% in non-German populations [12].

Independent analysis of molecular genetic causes of 54 patients from Russia with Stargardt’s disease showed 5 patients (in 9% of cases) with the presence of this complex allele [14].

In our study, the vast majority of cases with complex alleles were represented by the c.5882G>A, p.(Gly1961Glu) variant in trans (in 11 patients, giving allelic frequency 5.7%). Earlier, in a group of 21 patients from the Russian population with *ABCA4*-associated retinal pathology, it was determined that the combination of c.[1622T>C;3113C>T] and c.5882G>A mutations in 2 patients established a mild degree of disease [15].

The combination of the complex allele c.[1622T>C;3113C>T] and the genetic variant c.5882G>A in a compound heterozygous state was described in the population of two different regions of Germany (Heidelberg and Münster): 144 unrelated patients with STGD1, 200 unrelated patients with ARMD2, and 220 control subjects. In the group with STGD1 (166 patient chromosomes), three alleles with high frequency were found in the German population: the c.5882G>A allele with frequency 20.5% (34/166), the complex allele c.[1622T>C;3113C>T] with frequency 12.7% (21/166), and the c.2588G>C allele with frequency 10.2% (17/166). Based on the relatively high frequency of the complex allele in Germany, the authors hypothesized it has a German founder effect. It was mainly found in the compound-heterozygous state with the genetic variant c.5882G>A. No clinical–genetic correlations were found in this study (age of debut ranged from 9–25 years) [13].

In recent work on a group of 71 patients with different clinical and genetic variants of *ABCA4*-associated retinopathy using genotype–phenotype correlations, significant associations were found between genotype and the size of atrophic lesions, yellow spot volumes, and their corresponding rate of progression. The c.5882G>A allele of *ABCA4* has been shown to be associated with small lesions showing minimal spread over time: hypomorphic alleles, with preserved macular volumes due to foveal preservation, and severe biallelic variants, with extensive posterior pole atrophy and macular thinning. These are useful for patient prognosis and require genotype consideration in the design of future clinical trials [9]. In our study, the c.[1622T>C;3113C>T];[5882G>A] genotype demonstrated a significantly higher degree of disease manifestation, which corresponded to later onset of the disease compared to the c.[1622T>C];[3113C>T] compound heterozygous genotype.

In a research study from the United Arab Emirates, a novel complex allele, c.[2570T>C];[5882G>A], was identified. It determines the development of cone-rod dystrophy and is associated with a founder effect for the United Arab Emirates (100% in this case series). Homozygosity for this new complex allele causes cone-rod dystrophy in childhood but not macular dystrophy in young adulthood, which is associated with the c.5882G>A allele alone [8].

Our study not only expands the genetic landscape of *ABCA4* variants in patients with inherited retinal disease but also reveals novel associations of the particular genotype, with specific clinical progression of the retinal disease.

## 4. Materials and Methods

The analysis encompassed ninety-six probands from distinct unrelated families, all of whom exhibited confirmed *ABCA4*-related retinal pathology. The age spectrum of the patients spanned from 8 to 66 years, with a mean age of 22.63 years (standard deviation 11.35). The median of the age was 19.00 with an interquartile range of 14.00–29.25. A comprehensive approach was employed for patient assessment, involving medical specialists from various disciplines, such as genetics, ophthalmology, and neurology. The primary aim was to rule out hereditary syndromes and conditions affecting the eyes. This was facilitated by collecting detailed medical histories. Furthermore, all patients underwent standard ophthalmologic evaluations. Additionally, spectral optical coherence tomography and ocular fundus autofluorescence (FOCT) assessments were conducted, utilizing Zeiss Cirrus-5000 angiotomography. Electroretinography was conducted using diverse techniques, including the Ganzfeld electroretinogram, high-frequency rhythmic ERG at 30 Hz, and macular chromatic ERG employing a red stimulus through the Tomey EP-1000 system.

For the analysis of genotype–phenotype correlations, the subsequent clinical symptoms associated with *ABCA4*-related pathology were identified: the extent of disease manifestation, the pace of disease progression, the extent of impairment in best corrected visual acuity following WHO guidelines, the degree of color vision impairment, the presence of lens opacity, the degree of macular ERG abnormality, the level of macular area defect (as per OCT data), and the extent of pigment epithelium preservation. Clinical symptoms were ranged according to the scale summarized in Table 2.

Molecular genetic study was performed using paired-ended NGS on an Illumina MiSeq sequencer with the MiSeq reagent kit v2, using custom gene panel covered coding sequences of 212 genes. Ultra-multiplex PCR technology coupled with subsequent sequencing (AmpliSeq™) was used for library preparation. Subsequently, Sanger sequencing was performed for every variant in a proband and parents for segregation analysis.

Statistical analysis was performed in WinPEPI v. 11.65 program using the Fisher exact test [16].

## Figures and Tables

**Figure 1 ijms-24-16231-f001:**
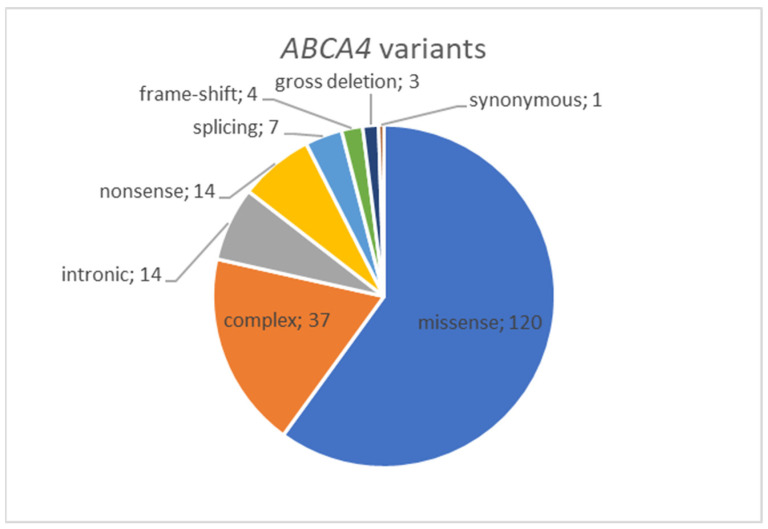
Distribution of ABCA4 variants by type revealed in Russian cohort of IRD patients.

**Figure 2 ijms-24-16231-f002:**
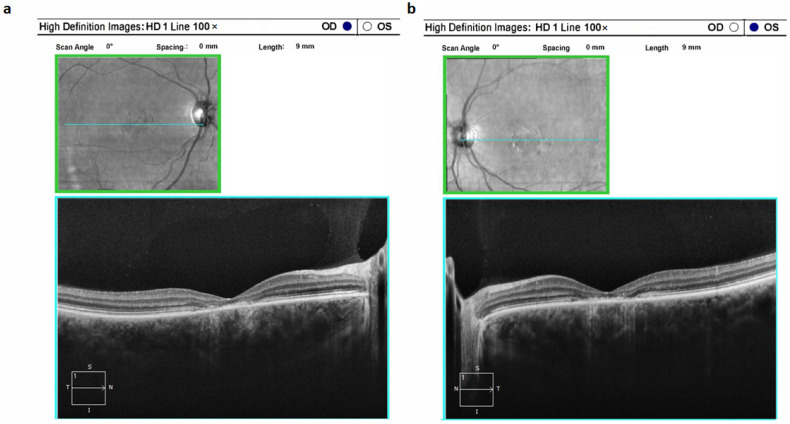
Optical coherence tomograms of the right (**a**) and left (**b**) eyes in a patient (ID-16) with Stargardt’s disease 1 (STGD1).

**Figure 3 ijms-24-16231-f003:**
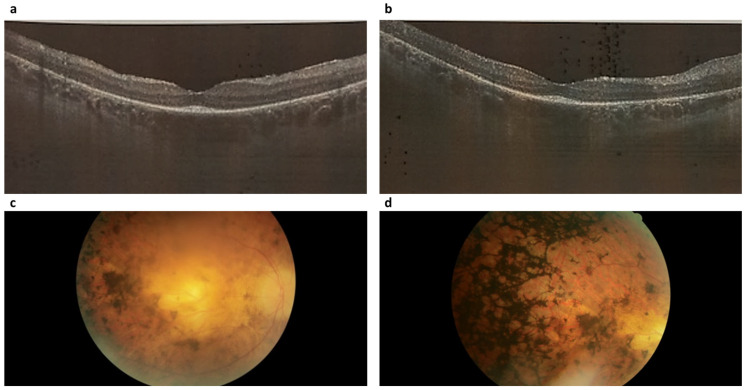
Optical coherence tomograms and image of the fundus of the right (**a**,**c**) and left (**b**,**d**) eyes of a patient (ID-76) with Retinitis pigmentosa 19 (RP19).

**Figure 4 ijms-24-16231-f004:**
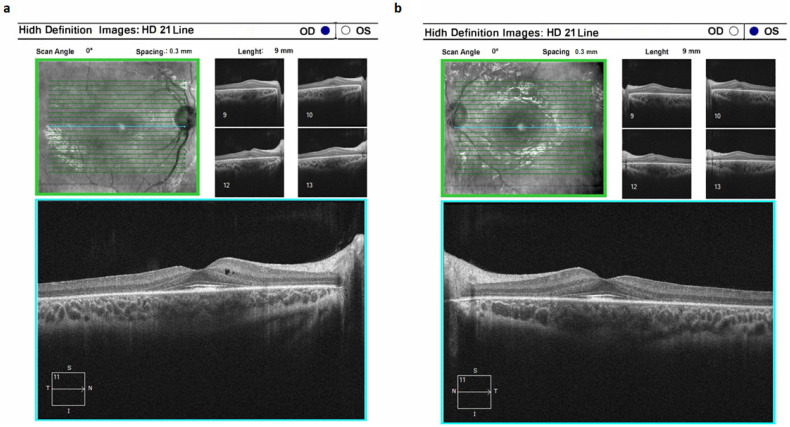
Optical coherence tomograms of the right (**a**) and left (**b**) eyes in a patient (ID-31) with cone-rod dystrophy 3 (CORD3).

**Table 1 ijms-24-16231-t001:** Contingency table of patients’ distribution into clinical groups with different types of mutations in *ABCA4* gene and probability values of Fisher’s exact test. LoF—loss-of-function allele.

Genotype	STGD1	Other Clinical Form	*p*-Value
Presence of at least one LoF allele in genotype	17	6	0.021
Non-LoF allele in genotype	68	5	

**Table 2 ijms-24-16231-t002:** Ranking of clinical symptoms in *ABCA4*-related retinal pathology for correlation.

Clinical Sign Degree	Meaning
Degree of disease manifestation	
1	under 5 years old
2	6–12 years old
3	13–18 years old
4	above 18 years old
Rate of disease progression	
1	6 months
2	6–12 months
3	>12 months
Degree of impairment of best corrected visual acuity	
1	Mild (0.67–0.33)
2	Moderate (0.28–0.125)
3	Severe (0.1–0.05)
4	Gross (0.04–0.02)
5	Complete (<0.02)
6	blindness
Degree of color vision impairment	
1	only the contrast perception of colors is impaired
2	lack of perception of 1–2 colors
3	achromatopsia
Presence of lens opacity	
1	No
2	Yes
Degree of macular ERG abnormality	
1	Normal
2	Subnormal
3	Unrecorded
Degree of macular area defect	
1	single defects
2	partial disintegration of layers
3	complete disintegration of layers
Degree of preservation of RPE	
1	partial destruction only in the macular area
2	absent only in the macular area
3	absent in the macular and paramacular area
4	undifferentiated to the extreme periphery

## Data Availability

The datasets used and/or analyzed during the current study are available from the corresponding author upon reasonable request.

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
