# Peer review of "Major Contribution of c.[1622T>C;3113C>T] Complex Allele and c.5882G>A Variant in ABCA4-Related Retinal Dystrophy in an Eastern European Population"

_ijms, 2023, doi:10.3390/ijms242216231_

Round 1
Reviewer 1 Report
Comments and Suggestions for Authors
Review on the manuscript “Major Contribution of c.[1622T>C;3113C>T] Complex Allele and c.5882G>A Variant in ABCA4-related Retinal Dystrophy in 3 an EASTERN EUROPEAN Population” by Kadyshev et al.
The authors presented data on the most common complex allele in ABCA4 gene known as a cause of several inherited retinal degenerations in Eastern European population. The genotype-phenotype correlation was also addressed. Interestingly, the authors showed that loss-of-function variants in ABCA4 associated with phenotypes distinct from Stargardt disease.
The reported here complex novel ABCA4 allele is not novel finding but it is common among patients from European part of Russian Federation as shown in this study. The manuscript is relatively well written though poorly structured and not well presented. Below, a list of comments that might be considered by the authors.
Through the entire manuscript use Italic for the ABCA4 gene.
Line 9 – inherited retinal diseses (IRD) is more frequently used than HRD
Line 24 –OMIM citation should be replaced with recent review or RetNet
Line 45 – provide reference or a link to ABCA4 variants database on “over 2000 genetic variants”. The number is currently is nearly 3000 in HGMD
Line 48 – consider revision of “clinical-genetic correlations”
Line 52 – consider revision “within the context of the Russian Federation”
Line 54- present results in the Table or diagram form
Line 62 – consider replacement of “the invariant dinucleotide region” by “canonical splice site ±2bp”
Line 63 – replace “solitary”
Line 78 – replace “the principal driver”
Line 87 – replace “instances”
Figure1 and 2 – remove test in Russian and provide proper figure legends
Line 117 – Consider “Genotype-phenotype correlations in ABCA4-related IRD/HRD”
Table 1 is not well presented– no explanation of any numbers, no comments or notes. Consider re-structuring, moving the Table 3 from the discussion to the results
Comments on the Quality of English LanguageEnglish should be improved. I would recommend proper service for better quality of scientific English.
Author Response
Comments and Suggestions for Authors
Review on the manuscript “Major Contribution of c.[1622T>C;3113C>T] Complex Allele and c.5882G>A Variant in ABCA4-related Retinal Dystrophy in 3 an EASTERN EUROPEAN Population” by Kadyshev et al.
The authors presented data on the most common complex allele in ABCA4 gene known as a cause of several inherited retinal degenerations in Eastern European population. The genotype-phenotype correlation was also addressed. Interestingly, the authors showed that loss-of-function variants in ABCA4 associated with phenotypes distinct from Stargardt disease.
The reported here complex novel ABCA4 allele is not novel finding but it is common among patients from European part of Russian Federation as shown in this study. The manuscript is relatively well written though poorly structured and not well presented. Below, a list of comments that might be considered by the authors.
Q1.1 Through the entire manuscript use Italic for the ABCA4 gene. – A.M.
A1.1 Thank you. We have changed the text accordingly.
Q1.2 Line 9 – inherited retinal diseses (IRD) is more frequently used than HRD – A.M.
A1.2 Thank you. We have changed the text accordingly.
Q1.3 Line 24 –OMIM citation should be replaced with recent review or RetNet– A.M.
A1.3 Thank you. We have updated citations.
Q1.4 Line 45 – provide reference or a link to ABCA4 variants database on “over 2000 genetic variants”. The number is currently is nearly 3000 in HGMD – A.M.
A1.4 We have access to the HGMD v.2022.1 where 1757 variants are collected. So, we made this reference within the text.
Q1.5 Line 48 – consider revision of “clinical-genetic correlations” – A.M.
A1.4 Thank you. We have replaced these words with “genotype-phenotype correlation”
Q1.6 Line 52 – consider revision “within the context of the Russian Federation” – A.M.
A1.6 We have rewrite this phrase.
Q1.7 Line 54- present results in the Table or diagram form
A1.7 We have added the diagram of different types of ABCA4 variants revealed in the cohort of IRD patients.
Q1.8 Line 62 – consider replacement of “the invariant dinucleotide region” by “canonical splice site ±2bp” – A.M.
A1.8 We have changed the text accordingly.
Q1.9 Line 63 – replace “solitary” – A.M.
A1.9 We have replace this word. Thank you.
Q1.10 Line 78 – replace “the principal driver” – A.M.
A1.10 Thank you. We have changed this word.
Q1.11 Line 87 – replace “instances” – A.M.
A1.11 Thank you. We replace this word throughout the text.
Q1.12 Figure1 and 2 – remove test in Russian and provide proper figure legends
A1.12 The figures were renumbered. Figure 3 (former 2) was replaced since there was a mistake. Russian text was removed and changed to English.
Q1.13 Line 117 – Consider “Genotype-phenotype correlations in ABCA4-related IRD/HRD” – A.M.
A1.13 Thank you for your suggestion. We have changed the title of section accordingly.
Q1.14 Table 1 is not well presented– no explanation of any numbers, no comments or notes. Consider re-structuring, moving the Table 3 from the discussion to the results – A.M.
A1.14 We have removed the Table 1 since the only statistically significant result is presented in the text. Table 2 (former 3) is a part of Materials and Methods describing scoring of patients’ symptoms.
Q1.15 Comments on the Quality of English Language
English should be improved. I would recommend proper service for better quality of scientific English.
A1.15 Thank you for your comment. We have revised the English language throughout the text.
Reviewer 2 Report
Comments and Suggestions for Authors
The authors present a cross-sectional study trying to correlate genotype and phenotype in patients with mutations in ABCA4 gene. This is one of the most frequent mutations leading to inherited retinal dystrophies, and analyzing possible relations between genotype and phenotype is scientifically interesting.
The manuscript is correctly presented, but there are some issues to revise before publication:
1) Affiliations should be complete for all authors.
2) Methods. They should appear between introduction and results.
4) Page 5. It should be explained what a 'LoF allele' is.
5) Methods. Page 7. Patients' age ranged from 8 to 66 years. What was the standard deviation?
6) Methods. Patient's age ranged from 8 to 66 years. Was this taken into account in the statistical analysis? A 10 year-old patient with Stargardt's disease may show a mild form, whereas a 60 year-old patient should be expected to show large retinal atrophy, that is, a very advanced stage of the disease. However, the latter could have started with ophthalmological symptoms or signs after 20 years of age. Therefore, I suppose that the study is biased because clinical severity assessment is not adequate (impairment of visual acuity, color vision impairment, lens opacity, macular area defect, and degree of preservation of RPE).
Author Response
The authors present a cross-sectional study trying to correlate genotype and phenotype in patients with mutations in ABCA4 gene. This is one of the most frequent mutations leading to inherited retinal dystrophies, and analyzing possible relations between genotype and phenotype is scientifically interesting.
The manuscript is correctly presented, but there are some issues to revise before publication:
Q2.1 1) Affiliations should be complete for all authors. – A.M.
A2.1 All the authors represent one affiliation. We added index to point that out.
Q2.2 2) Methods. They should appear between introduction and results. – A.M.
A2.2 According to the requirement of International Journal of Molecular Sciences, Materials and Methods section is placed after the Discussion. We followed the requirements of the Journal in the manuscript structure.
Q2.3 4) Page 5. It should be explained what a 'LoF allele' is. – A.M.
A2.3 We have decipher this abbreviation here, thank you.
Q2.4 5) Methods. Page 7. Patients' age ranged from 8 to 66 years. What was the standard deviation?
A2.4 We updated Supplementary Table 1 with the information about age. We also updated the M&M section with analysis of this sample.
Q2.5 6) Methods. Patient's age ranged from 8 to 66 years. Was this taken into account in the statistical analysis? A 10 year-old patient with Stargardt's disease may show a mild form, whereas a 60 year-old patient should be expected to show large retinal atrophy, that is, a very advanced stage of the disease. However, the latter could have started with ophthalmological symptoms or signs after 20 years of age. Therefore, I suppose that the study is biased because clinical severity assessment is not adequate (impairment of visual acuity, color vision impairment, lens opacity, macular area defect, and degree of preservation of RPE).
Q2.6 Thank you for your question. The age of the patient at the time of inclusion in the study was indeed variable from 8 to 66 years, but the age of manifestation (debut) of the disease in most cases does not coincide with the age at the time of examination of the patient and establishment of the clinical and genetic diagnosis. The study included a mandatory retrospective study of medical records and data of instrumental examination of the patient at an earlier (childhood) age, as a result, the registered data on the disease debut, rather than the age of the patient at the time of the present study, were taken into account when assessing the criterion of the age of manifestation. This was necessarily taken into account in the statistical analysis. Indeed, the older the patient gets, the more severe the disease is, that is why the data of instrumental examination (optical coherence tomography, perimetry, visual acuity, electroretinography data) in dynamics were analyzed. This is an extremely important component of the study, as it is incorrect to establish the age of the pathology debut only on the basis of the data obtained at the time of this study. Taking into account the localization of ABCA4 gene expression and clinical variability, the pathological process may debut from both pigment epithelium and photoreceptor discs, or simultaneously from both components, due to which a different clinical portrait develops and a different clinical diagnosis is formed. In view of these facts, different degrees of damage of certain retinal layers (cone and rod apparatus, pigment epithelium, outer boundary membrane) are recorded and, as a consequence, corresponding changes in functional vision parameters are registered, including color perception and maximum corrected visual acuity.
We have updated the Results Section with these considerations.
Round 2
Reviewer 1 Report
Comments and Suggestions for Authors
The authors' reply is satisfactory
Comments on the Quality of English LanguageThere are
Reviewer 2 Report
Comments and Suggestions for Authors
Changes have been performed. No further issues detected.